# Water Intake and Handgrip Strength in US Adults: A Cross-Sectional Study Based on NHANES 2011–2014 Data

**DOI:** 10.3390/nu15204477

**Published:** 2023-10-23

**Authors:** Dongzhe Wu, Chaoyi Qu, Peng Huang, Xue Geng, Jianhong Zhang, Yulin Shen, Zhijian Rao, Jiexiu Zhao

**Affiliations:** 1Exercise Biological Center, China Institute of Sport Science, Beijing 100061, China; wdz276640188@outlook.com (D.W.); 15201758783@163.com (C.Q.); qdqhuangpeng@163.com (P.H.); 18896556283@163.com (X.G.); shenyulin@ciss.cn (Y.S.); raoz19@shnu.edu.cn (Z.R.); 2Department of Exercise Physiology, Beijing Sport University, Beijing 100084, China; 3School of Exercise and Health, Shanghai University of Sport, Shanghai 200438, China; 4National Institute of Sports Medicine, Beijing 100763, China; zjhong0706@163.com; 5College of Physical Education, Shanghai Normal University, Shanghai 200233, China

**Keywords:** daily total intake of water, handgrip strength, National Health and Nutrition Examination Survey

## Abstract

This study aimed to examine the relationship between daily total intake of water (DTIW) and handgrip strength (HGS) among US adults and to explore the impact of water intake on muscle function and health, providing a reference for public health policies and health education. Using the data from the National Health and Nutrition Examination Survey (NHANES) 2011–2014, a cross-sectional survey design was adopted to analyze 5427 adults (48.37% female and 51.63% male) aged 20 years and above. DTIW was assessed using two non-consecutive 24 h dietary recall interviews, and the HGS level was measured using a Takei Dynamometer. Weighted generalized linear regression models and restricted cubic spline plots were used to analyze the linear and nonlinear associations between DTIW and HGS level and to conduct a gender subgroup analysis and an interaction effect test. The results showed that there were significant differences in HGS and other characteristics among different quartile groups of DTIW (*p* < 0.05). There was a significant nonlinear trend (exhibiting an inverted U-curve) between DTIW and HGS (*p* for nonlinear = 0.0044), with a cut-off point of 2663 g/day. Gender subgroup analysis showed that the nonlinear trend (exhibiting an inverted U-curve) was significant only in males (*p* for nonlinear = 0.0016), with a cut-off point of 2595 g/day. None of the stratified variables had an interaction effect on the association between DTIW and HGS (*p* for interaction > 0.05). In conclusion, this study found a nonlinear association between DTIW and HGS levels, as well as a gender difference. This finding provides new clues and directions for exploring the mechanism of the impact of DTIW on muscle function and health and also provides new evidence and suggestions for adults to adjust their water intake reasonably.

## 1. Introduction

Water is an essential nutrient for normal physiological and metabolic processes of the human body, and it is vital for the normal function maintenance of human organs and systems [1,2]. In addition to serving as a medium for digestion, absorption, metabolism, and excretion of metabolic products, body water is also involved in important physiological processes such as tissue structure and function and body temperature balance regulation. Long-term dehydration can cause various health problems [3,4,5,6] (increased mortality, chronic kidney disease, gallstones, coronary heart disease, etc.), and even a loss of about 1% or 2% of body water can impair cognitive function and exercise capacity [2].

Water is the main component of the body, accounting for about 76% of muscle mass [7]. As we age, there is a notable reduction in total body water (TBW) and intracellular water (ICW) [8]. This decline in hydration status not only predisposes the elderly to various health risks but also potentially exacerbates age-related decrements in muscle mass and functional strength, engendering a complex interplay among aging, hydration, and muscular function [9]. Lorenzo et al. [7] reviewed that the partial loss of ICW is associated with the reduction of muscle mass or indirectly induced with insufficient cellular hydration (the structure and enzyme activity of intracellular proteins are affected by water). Muscle, as the main water storage depot of the body, will act as the main organ to lose water first when the body is exposed to dehydration or an insufficient hydration physiological state to protect the normal operation of important physiological organs such as the brain and liver.

In different mammalian cell types, cell volume plays a crucial role in insulin-mediated action, and skeletal muscle, as the main site of insulin action, also highlights the importance of adjusting its intracellular water balance. Insufficient hydration or dehydration will have a significant negative impact on the mechanical and metabolic functions of skeletal muscle [10]. In terms of energy metabolism, muscle is responsible for most glucose metabolism and plays an important role in the formation and treatment of insulin resistance and type 2 diabetes. Insulin mainly works by promoting glucose uptake with liver and muscle cells. Valentin et al. [11] further confirmed that the ratio of glycogen reserves to water reserves in skeletal muscle is 1:3, and the recovery of glycogen and water in skeletal muscle is a synergistic process after exercise. When dehydration occurs in muscle cells, it may cause muscle glycogen synthesis disorder, which further affects muscle mass/strength; however, more studies are needed to confirm this conclusion. Previous studies have confirmed that there is a potential association between hydration status and muscle strength/mass in individuals. Joana et al. [12] investigated the relationship between hydration status and handgrip strength in elderly people and found that insufficient hydration in women was significantly associated with lower handgrip strength. Yoo et al. [13] conducted a national cross-sectional survey study and found that insufficient water intake in elderly people will increase the risk of sarcopenia.

In summary, the mechanical and metabolic functions of skeletal muscle will have a certain degree of a negative impact when there is insufficient hydration or dehydration. Previous studies have predominantly concentrated on the relationship between hydration status and muscle mass, whereas it is essential to highlight that the bulk of muscle strength decline cannot be solely attributed to the reduction in muscle mass. This implies the potential existence of other factors that may exert a more substantial influence on muscle strength than mere changes in muscle mass [14,15]. The purpose of this study is to investigate the association between daily total water intake and handgrip strength in adults, utilizing extensive nationwide cross-sectional survey data from the National Health and Nutrition Examination Survey (NHANES). This research aims to provide scientific reference data for the field of human dietary nutrition and health while shedding light on the intricate interplay between hydration and muscle function. Our study posits the existence of a potential linear or nonlinear relationship between daily total water intake (DTIW) and handgrip strength (HGS) among the adult population, with the possibility of gender-based variations. Specifically, it suggests that hydration status may exacerbate age-related declines in muscle mass and functional strength.

## 2. Materials and Methods

### 2.1. Study Subjects and Data Sources

This study used data from the 2011–2014 National Health and Nutrition Examination Survey (NHANES), a nationally representative survey conducted by the Centers for Disease Control and Prevention (CDC), to assess the health and nutrition status of American adults and children. This study’s participants were sourced from the NHANES database by the Strengthening the Reporting of Observational Studies in Epidemiology (STROBE) guidelines for cross-sectional studies. The survey was carried out by the National Center for Health Statistics and the CDC every two years, using a multi-stage probability sampling design to examine about 10,000 non-institutionalized individuals from across the United States. Data collection included household interviews and physical examinations. During the interviews, participants answered questions on demographic, socioeconomic, dietary, and health-related variables, while the examinations measured medical, dental, and physiological indicators.

From 2011 to 2014, the NHANES sampled 18,591 participants, of whom 10,907 were adults (≥20 years old). We collected 10,021 valid individual interview questionnaires (excluding 886 invalid ones). Among them, 9895 participants underwent physical examinations (excluding 126 non-participants); 7390 participants completed physical activity questionnaires (excluding 2505 non-participants); 6070 participants underwent dietary recall interviews (excluding 1320 non-participants); 5770 participants completed smoking and alcohol consumption questionnaires (excluding 300 non-participants); 5707 participants underwent comorbidity surveys for diabetes, hypertension, and hyperlipidemia (excluding 63 non-participants); and 5427 participants underwent handgrip strength tests (excluding 280 non-participants). Finally, we included and retained valid data from 5427 participants in this study (the detailed screening process is shown in Figure 1). The survey was conducted in accordance with the Declaration of Helsinki.

### 2.2. Dietary Assessment

During the NHANES survey period, professional investigators collected and assessed the average of two separate 24 h dietary recall interviews. The first recall was collected in person at the mobile examination center, and the second interview was conducted via a telephone consultation 3–10 days later. Using two non-consecutive days of dietary intake data was more accurate than using single-day data. Participants were asked to recall the details of food and beverage consumption in the previous 24 h, and corresponding nutritional intake analysis was performed according to professional personnel’s report on dietary intake. The nutritional intake indicators included in this study were all calculated as the average of two non-consecutive 24 h dietary recalls.

### 2.3. Daily Total Intake of Water

In the NHANES study, a face-to-face 24 h dietary recall interview was conducted with each participant during their visit to the Mobile Examination Center (MEC) within a dedicated dietary interview room. This room was equipped with essential tools, including a computer equipped with the United States Department of Agriculture (USDA) Automated Multiple Pass software (https://wwwn.cdc.gov/Nchs/Nhanes/2013-2014/DR1TOT_H.htm) access date September 2016, food models, and various three-dimensional measuring instruments such as glasses, bowls, mugs, measuring mounds, circles, thickness sticks, spoons, rulers, cartons, and water bottles of different sizes. MEC interviewers, who had received specialized training, elucidated the purpose of the dietary recall, elucidated the interview process in detail, and presented standardized questions to each participant in a completely unbiased manner. Initially, participants were instructed to retrospectively recall all foods and beverages consumed during the preceding 24 h period (from midnight to midnight), encompassing items consumed both at home and away from home, including snacks, coffee, soft drinks, water, and alcoholic beverages. Specific inquiries were performed regarding brand details, preparation methods, and quantities consumed, including queries about food additives, such as milk added to cereal or coffee. Food models and measuring tools were employed to assist participants in estimating portion sizes [16].

Regarding water intake, participants were questioned about their consumption of tap water, including filtered tap water and water from drinking fountains. Additionally, various brands of bottled waters (plain, spring, mineral, and electrolyte-fortified) and carbonated plain waters (sparkling, seltzer, and club soda) were considered. Water intake also encompassed liquids added to food and beverages, such as various types of liquid milk, fruit juice, vegetable juice, juice drinks, carbonated and non-carbonated sugared beverages, coffee, tea, hot chocolate, and alcoholic beverages [17,18]. This intake was calculated by accumulating water content per milliliter of each beverage. Ice was also accounted for in the recording process. As participants reported each food and beverage, MEC interviewers entered the data into the USDA Automated Multiple Pass software, cross referencing it with entries in the USDA nutrient composition database. The software consolidated the entered data and generated estimates for total daily nutrient intake, including category-specific nutrient intake such as water, protein, carbohydrates, and other nutrients, which were utilized in our analysis.

### 2.4. Handgrip Strength

In this study, handgrip strength was used as a dependent variable, measured in kg. Handgrip strength (HGS) was measured using the Takei Dynamometer (TKK 5401; Takei Scientific Instruments, Tokyo, Japan) for adults aged 20 years and older. It is worth mentioning that handgrip strength measurements were only available for the years 2011 to 2014, as this was the period during which the handgrip tests were conducted as part of the survey.

Subjects were asked to maintain an upright posture, position arm vertically downward, handgrip the dynamometer for the strength test, repeat the test three times with both hands (dominant hand, non-dominant hand), with a 60 s interval between each measurement, and the sum of the average of the highest peak handgrip strength of both hands was taken as the maximum absolute handgrip strength. To further improve the objectivity of this study, relative handgrip strength was used for subsequent analysis in this study [19,20].
Relative maximal handgrip strength = maximal handgrip strength (kg)/weight (kg)

### 2.5. Covariate

In terms of the rationale for covariate selection, the scientific rigor of our study is ensured through a process of logical sorting and screening. Primary demographic characteristics considered include age [21], gender [22,23], BMI [24], and educational level [25] due to their significant correlations with muscular strength. Additionally, research has indicated a positive correlation between higher income and superior performance across multiple domains of physical function [26]. Smoking and alcohol consumption have demonstrated to exert noticeable negative effects on muscular composition and strength in both human [27,28] and animal studies [29,30]. Moreover, scientific evidence unequivocally underscores the beneficial impact of systematic physical activity on enhancing human health and muscular strength levels [28,31,32]. Furthermore, the intake of scientific nutrients, including protein [33], fat [34,35], carbohydrates [36,37], energy [38], and sugar [39,40], plays a pivotal role in muscular development. Existing studies have affirmed that common societal chronic conditions, such as hypertension [41,42], hyperlipidemia [43,44,45], diabetes [46,47,48], and cancer [49], operate through distinct physiological pathways to mediate the decline in muscular strength and muscle mass.

Demographic factors: age, gender (male, female), race (Mexican American, non-Hispanic black, non-Hispanic white, other race), education level (below: less than high school; high school; above: more than high school), and body mass index (<25, 25–29.9, ≥30 kg/m^2^).

Socioeconomic status: poverty/income ratio, calculated by dividing household (or individual) income by the poverty guideline for the survey year (low-income PIR ≤ 1.3, medium income 1.3 < PIR < 3.5, high income ≥ 3.5).

Dietary nutrient intake: daily total intake of protein, fat, carbohydrates, energy, and sugar recorded with two non-consecutive 24 h dietary recall interviews and calculated based on dietary nutrient content.

Lifestyle habits: smoking status (never: smoked less than 100 cigarettes in their lifetime; former: smoked more than 100 cigarettes in their lifetime, now not at all; current: smoked more than 100 cigarettes in their lifetime, some days or every day), alcohol consumption (never: <12 drinks in their lifetime; former: ≥1 drink in the past 12 years, none in the past year, or none in the past year but ≥12 drinks in their lifetime; mild: ≤1 drink per day if female, ≤2 drinks per if male; moderate: ≤2 drinks per day if female, ≤3 drinks per day if male; and heavy: ≤3 drinks per day if female, ≤4 drinks per day if male), physical activity (time and energy expenditure of typical physical activity in the past week, including vigorous and moderate physical activity at work, commuting, and leisure time), measured using the metabolic equivalent (weekly metabolic equivalent score, METs h/week).

Health conditions: hypertension diagnosis (diagnosed by a doctor or health professional? Ever used antihypertensive drugs? Systolic blood pressure ≥ 140 mmHg and diastolic blood pressure ≥ 90 mmHg in three blood pressure measurements), hyperlipidemia diagnosis (triglycerides (TG) ≥ 150 mg/dL; serum total cholesterol (TC) ≥ 200 mg/dL; low-density lipoprotein (LDL) ≥ 130 mg/dL; high-density lipoprotein (HDL) < 40 mg/dL (male), <50 mg/dL (female); use of lipid-lowering drugs), diabetes diagnosis (diagnosed by a doctor or health professional? Glycated hemoglobin (HbA1c) ≥ 6.5 mmol/L; fasting blood glucose (GHLU) ≥ 7.0 mmol/L; ever used anti-diabetic drugs?), cancer diagnosis (diagnosed by a doctor or health professional?).

### 2.6. Statistical Methods

This study followed the NHANES complex sampling survey procedure and used the complex sampling weights provided by the NHANES analysis guide. Weighted data were used for indicator analysis to produce nationally representative data estimates. Continuous variables involved in this study were expressed as means (standard error), and categorical variables were expressed with the actual number (weighted percentage). Group difference test: one-way analysis of variance was used for continuous variables, and chi-squared test was used for categorical variables. A weighted generalized linear regression model was used to analyze the linear relationship between total water intake (quartile group) and handgrip strength, and further gender subgroup analysis was performed. In addition, based on the linear regression results, non-restrictive cubic graphs were used to test the nonlinear trend between variables. Finally, the control variables were included in the interaction effect test model to further verify whether there was an interaction effect between total water intake and handgrip strength in comparison to the control variables. The two-sided *p* < 0.05 was used as the threshold for statistical difference. All analyses were performed using R Studio (4.2.1, USA).

## 3. Results

### 3.1. Baseline Characteristics of the Study Population

Table 1 presents the characteristics of this study’s population grouped with DTIW quartiles in NHANES 2011–2014. A total of 5427 valid data were included in this study (48.37% female and 51.63% male), and DTIW was divided into Quartile 1, Quartile 2, Quartile 3, and Quartile 4 groups. There were significant differences in handgrip strength, gender, age, race, education level, poverty/income ratio, smoking status, alcohol consumption, physical activity level, daily total intake of protein, daily total intake of energy, daily total intake of carbohydrates, daily total intake of fat, and hyperlipidemia among different quartile groups (*p* < 0.05), while there were no group differences in body mass index, daily total intake of sugars, diabetes, hypertension, and cancer (*p* > 0.05).

This study unveiled the associations between various multifaceted factors and daily total intake water (DTIW). Individuals who were male or possessed higher education levels exhibited higher water intake, while water intake gradually decreased in adults aged over 60. Compared to other races, water intake was the highest among the white population and the lowest among the black population. Economically, an upward trend in water intake was observed with the enhancement of personal economic status. Non-smokers and mild drinkers exhibited the highest water intake, displaying a positive association between healthy lifestyle choices and adequate water intake. From the perspective of nutritional intake and physical activity level, a positive correlation trend with water intake was noticed, suggesting that physical activity and appropriate nutritional intake might promote higher water intake. Compared to the healthy population, individuals with hyperlipidemia had higher water intake, and, concurrently, an upward trend in water intake was observed with the increase in the population’s handgrip strength level. These findings provided insights for social health and clinical practice, emphasizing the significance of considering multifaceted factors at both individual and community levels in public health strategies to promote adequate water intake and overall health.

### 3.2. Association Analysis between Daily Total Intake of Water and Handgrip Strength in American Adults

Table 2 showed the weighted generalized linear regression model of DTIW and HGS. In the Crude Model, when compared with the lowest quartile of DTIW, a more significant positive correlation was observed between DTIW Q3 (β = 0.03, 0.01~0.05, *p* = 0.01) and Q4 (β = 0.07, 0.04~0.09, *p* < 0.0001) and HGS, with a high level of significance found in the trend test (*p* < 0.0001). In the fully adjusted model (Model 2), there was no significant correlation observed between DTIW Q2 (β = 0.01, −0.01~0.02, *p* = 0.48), Q3 (β = 0, −0.02~0.02, *p* = 0.87), Q4 (β = 0.01, −0.00~0.03, *p* = 0.15) and HGS, and no significance was found in the trend test (*p* > 0.05).

Table 3 showed the results of the gender subgroup weighted generalized linear regression model of DTIW and HGS. In the fully adjusted model for males (Model 2), when compared with the lowest quartile of DTIW, no significant correlation was observed between DTIW Q2 (β = 0.01, −0.01~0.04, *p* = 0.31), Q3 (β = 0.02, −0.01~0.04, *p* = 0.27), Q4 (β = 0.02, −0.01~0.05, *p* = 0.13), and HGS; no significance was found in the trend test (*p* > 0.05). In the fully adjusted model for females (Model 2), when compared with the lowest quartile of DTIW, no significant correlation was observed between DTIW Q2 (β = 0, −0.02~0.01, *p* = 0.62), Q3 (β = −0.01, −0.03~0.01, *p* = 0.25), Q4 (β = 0, −0.02~0.02, *p* = 0.99), and HGS; no significance was found in the trend test (*p* > 0.05).

The potential nonlinear association between DTIW and HGS was further explored with a restricted cubic spline (RCS) graph. Figure 2 controlled the variables according to the fully adjusted linear regression model (Model 2) (covariates were age, gender, race, poverty/income ratio, education level, body mass index, physical activity level, smoking status, alcohol consumption, daily total intake of protein, daily total intake of energy, daily total intake of carbohydrates, daily total intake of sugars, daily total intake of fat, smoke status, diabetes, hypertension, hyperlipidemia, and cancer), and the results showed that there was a significant nonlinear trend between DTIW and HGS (*p* for nonlinear = 0.0044). The RCS graph showed that the cut-off point of HGS with the increase in DTIW was 2662.88 g/day, suggesting that when the daily total water intake was less than 2662.88 g, HGS increased with the rise in DTIW; on the contrary, when the daily total water intake was greater than 2662.88 g, HGS decreased with the increase in DTIW.

Figure 3 controlled the variables according to the fully adjusted linear regression model for males (Model 2) (covariates were age, race, poverty/income ratio, education level, body mass index, physical activity level, smoking status, alcohol consumption, daily total intake of protein, daily total intake of energy, daily total intake of carbohydrates, daily total intake of sugars, daily total intake of fat, smoke status, diabetes, hypertension, hyperlipidemia, and cancer), and the results showed that there was a significant nonlinear trend between DTIW and HGS in males (*p* for nonlinear = 0.0016). The RCS graph showed that the cut-off point of HGS in males with the increase in DTIW was 2595.08 g/day, suggesting that when the daily total water intake was less than 2595.08 g, males’ HGS increased with the rise in DTIW; on the contrary, when DTIW was greater than 2595.08 g, males’ HGS decreased with the increase in DTIW.

Figure 4 controlled the variables according to the fully adjusted linear regression model for females (Model 2) (covariates were age, race, poverty/income ratio, education level, body mass index, physical activity level, smoking status, alcohol consumption, daily total intake of protein, daily total intake of energy, daily total intake of carbohydrates, daily total intake of sugars, daily total intake of fat, smoke status, diabetes, hypertension, hyperlipidemia, and cancer), and the results showed that there was no significant nonlinear trend between DTIW and HGS in females (*p* for nonlinear = 0.6983).

### 3.3. Interaction Effect Test

We performed stratified regression analyses based on gender, age, race, body mass index, education level, poverty/income ratio, smoking status, alcohol consumption, physical activity level, daily total intake of protein, daily total intake of energy, daily total intake of carbohydrates, daily total intake of sugars, daily total intake of fat, diabetes, hypertension, hyperlipidemia, and cancer, and adjusted for these variables as covariates. As shown in Table 4, none of the selected stratification variables interacted with the association between DTIW and HGS (*p* for interaction > 0.05), indicating an independent nonlinear relationship between DTIW and HGS.

## 4. Discussion

This study used the NHANES data from 2011–2014 and used a generalized linear regression model and restricted cubic spline graph to explore the association between total water intake (DTIW) and handgrip strength (HGS) in American adults. The results showed that DTIW and HGS had a nonlinear association, and there was a turning point (about 2662.88 g/day of water intake) in this nonlinear association. When DTIW was lower than the turning point, HGS increased with the increase in DTIW; when DTIW was higher than the turning point, HGS decreased with the rise in DTIW. This nonlinear trend had gender differences, and the association was more significant in males than in females. The interaction effect test results further confirmed that gender, age, race, body mass index, education level, poverty/income ratio, smoking status, alcohol consumption, physical activity level, daily total intake of protein, daily total intake of energy, daily total intake of carbohydrates, daily total intake of sugars, daily total intake of fat, diabetes, hypertension, hyperlipidemia, and cancer did not have an interaction effect on the association between DTIW and HGS, which, to some extent, indicates that the nonlinear association between DTIW and HGS found in this study has strong stability and universality, and that it is not affected or moderated by other factors in this study. The results of this study suggest that appropriate water intake may be beneficial to maintain and improve handgrip strength level in adults. Still, excessive water intake may adversely affect handgrip strength level. The social practice application significance of this study provides new evidence and insights into the relationship between water intake and handgrip strength level in adults, which helps to guide the formulation and implementation of public health policies and health education.

Handgrip strength level is an important indicator reflecting muscle function and physical health status and is closely related to various health outcomes such as chronic diseases, disabilities, cognitive function, mental health, quality of life, and mortality risk [50,51,52,53,54]. Therefore, maintaining and improving handgrip strength level is significant for preventing and delaying aging and improving health and wellbeing. Water intake is an important factor affecting muscle strength/mass; however, the current research on the relationship between water intake and handgrip strength level is still scarce, and there are inconsistent results present between studies [55]. Hyeonmok et al. [55] found, in a cross-sectional study (based on the data of the Korean National Health and Nutrition Survey 2014–2015, including 1024 elderly people aged 65 and above), that HGS had a significant association with water intake in univariate analysis; however, when adjusting for age, BMI, strength training, and other covariates, the significant association disappeared, suggesting that water intake may not be the main factor affecting the handgrip strength of the elderly, but may be moderated by other factors such as age, body shape, and resistance training. This survey provides reference data and theoretical reference for this study. The generalized linear weighted regression results of HGS and DTIW in this study are relatively consistent with the above research; however, this study further explores its potential association by using an RCS-restricted cubic spline graph to verify its nonlinear association trend. Compared to previous studies, this research provides a more comprehensive examination of various potential variables affecting handgrip strength, such as physical activity, race, poverty/income ratio, diabetes, hypertension, hyperlipidemia, etc. It is worth noting that prior studies did not include dietary and beverage intake when considering hydration levels, which could compromise the objectivity of the results. This study also pays special attention to the impact of cancer on muscle strength. Specifically, cancer affects muscle strength through two primary mechanisms: first, by inducing cachexia, which leads to significant losses in body weight and muscle mass, thereby directly reducing muscle strength; and second, through the activation of the NF-κB signaling pathway [56]. This activation triggers inflammation and muscle degradation, further weakening muscle strength and disrupting the expression of Pax7, a critical factor for muscle regeneration. The cumulative effect of these factors leads to a significant decline in muscle strength among cancer patients.

Although the nonlinear trend test between DTIW and HGS in females was not significant (*p* for nonlinear = 0.6983) in this study, we found that the nonlinear trend direction was consistent with the nonlinear trend direction in males (both exhibited an inverted U-curve) according to Figure 3. It can be seen that insufficient or excessive water intake will also affect the development of female handgrip strength to some extent. Based on the relevant literature and theories, gender may be an important factor affecting water intake and muscle function [57,58], as there are some biological and sociocultural differences between men and women. These differences may lead to different needs, responses, and outcomes of water intake for them. As there are some differences between males and females in their biology and socio-culture, gender may be an important factor affecting water intake and muscle function, and these differences may lead to different needs, responses, and outcomes of water intake for them. Generally speaking, males need more water intake than females [59] because males have a higher body weight, body surface area, basal metabolic rate, and sweating rate. These factors require males to have a higher water consumption and need to replenish more water to maintain their body’s normal operation. At the same time, males also have stronger muscle function than females because males have more skeletal muscle mass, a higher testosterone level, and a higher growth hormone level [60,61,62]. These factors enable males to have increased muscle development, contraction, and recovery ability, and they need to use more water to provide energy, transport oxygen, and eliminate waste to their bodies. Therefore, at the same water intake level, males may be more susceptible to the positive or negative effects of water on muscle function in comparison to females.

The research on the association between water intake and muscle strength is still relatively scarce. Based on the results of this study and the previous literature, we intend to make reasonable assumptions and explanations from the aspects of intracellular water balance, blood circulation, nervous system function, etc.

Intracellular water balance refers to the osmotic pressure balance between intracellular fluid and extracellular fluid, which determines the volume and shape of cells [63]. The intracellular proteins in muscle cells include myosin and actin, which are the main components of muscle fibers and also the main factors for muscle contraction. Intracellular water balance is affected by the hydration status [1,64]. When the body is in a low-hydration state (or dehydration), the solute concentration in the extracellular fluid increases, causing the intracellular fluid to move to the extracellular fluid, making the cells shrink or shrivel; when the body is in a high-hydration state (or overhydration), the solute concentration in the extracellular fluid decreases, causing the extracellular fluid to move to the intracellular fluid, making the cells swell or dissolve. These changes can induce conformational changes or functional loss of intracellular proteins in muscle cells, thus affecting the interaction between myosin and actin [65] and then affecting the formation and detachment of the crossbridge–actin complex, reducing the contraction ability and strength of muscle fibers.

Blood circulation refers to the movement of blood through the vascular system between various organs and tissues in the whole body as a result of the function of the heart [66]. Blood circulation is also affected by the hydration status [67,68]. When the body is in a low-hydration state (or dehydration), the water content in the blood decreases, the solute concentration increases, the blood viscosity increases, the fluidity decreases, and the blood circulation is blocked; when the body is in a high-hydration state (or overhydration), the water content in the blood increases, the solute concentration decreases, the blood viscosity decreases, the fluidity increases, and the blood circulation is too fast. These changes will affect the delivery efficiency of nutrients and hormones in the blood, which include oxygen, glucose, amino acids, insulin, growth hormone, etc., which are essential factors for supporting muscle metabolism and synthesis [69,70]. When the delivery efficiency of nutrients and hormones in the blood is low, it will cause muscle cells to lack oxygen, glucose, amino acids, insulin, growth hormone, and other factors, thus affecting the metabolism and synthesis of muscle cells, reducing the energy production and protein increase in muscle cells.

Nervous function refers to the ability of neurons to communicate information with other cells through potential changes or neurotransmitter releases [71,72]. Nervous system function is also affected by the hydration status [73,74]. When the body is in a low-hydration state (or dehydration), the water and electrolyte imbalance inside and outside the nerve cells leads to an abnormal membrane potential and action potential of nerve cells and the synaptic transmission disorder between nerve cells [75] (indirectly leading to neurotransmitter release and binding obstruction, nerve signal transmission delay or distortion, nerve network stability and plasticity reduction, and other series of adverse consequences); when the body is in a high-hydration state (or overhydration), the water and electrolyte excess inside and outside the nerve cells leads to an abnormal membrane potential and action potential of nerve cells, and the synaptic transmission abnormality between the nerve cells (indirectly leading to neurotransmitter release and binding abnormality, nerve signal transmission which is too fast or chaotic, nerve network stability and plasticity reduction, and other series of adverse consequences). These changes will affect the excitability and conduction speed of motor neurons, which determine the speed and quality of motor commands from the brain to the muscles. When the excitability and conduction speed of motor neurons are low, it will cause a motor command transmission delay or distortion, thus affecting the coordination and strength of muscle contractions [76,77].

Based on the above theory, this study explored the mechanism of water intake on muscular strength based on the relevant theory. Appropriate water intake can maintain the balance of water inside and outside of the cell, promote smooth blood circulation, increase the delivery efficiency of nutrients and hormones, ensure the normal function of the nervous system, and enhance the excitability and conduction speed of motor neurons, thereby enhancing the contraction ability and strength of muscle fibers. On the contrary, excessive water intake will interfere with the balance of water inside and outside of the cell, cause blood circulation disorder, reduce the delivery efficiency of nutrients and hormones, damage the normal function of the nervous system, and reduce the excitability and conduction speed of motor neurons, thereby weakening the contraction ability and strength of muscle fibers, and further affect the metabolism, energy, and protein level in muscle cells, and induce muscle synthesis decline and muscle strength decline.

This study, based on the NHANES data, explored the nonlinear association between daily total water intake and handgrip strength level in adults, providing new clues and directions for the mechanism of water intake on muscle function and physical health, providing a new basis and suggestions for how adults can reasonably adjust water intake, providing further evidence and support for strengthening the attention and concern for the relationship between water intake and handgrip strength level in public health policies, and helping to promote the health, welfare, and economic benefits of individuals and society. This study provides valuable information and references for researchers, educators, practitioners, the public, and others within related fields. However, there are still some limitations in this study: (1) This study used the NHANES data, which is a cross-sectional study design, which cannot determine the causal relationship between total water intake and handgrip strength and can only reflect the correlation between them. Therefore, the results of this study cannot exclude the influence of other potential confounding factors or mediating factors and cannot accurately infer the long-term impact of the total water intake on handgrip strength. In order to verify the causal mechanism between total water intake and handgrip strength, it is necessary to conduct more randomized self-controlled experiments or longitudinal cohort studies. (2) This study used total water intake as an indicator of water intake, which includes the water from drinking water, beverages, and food. However, different sources of water may have different effects on handgrip strength levels. For example, sugary drinks may affect blood glucose and insulin levels, thus affecting muscle metabolism and synthesis; water in food may interact with other nutrients, thus affecting muscle function and health. It is important to note that the use of a dietary recall to assess water intake also represents a study limitation. Due to data source limitations, this study did not distinguish the effects of different sources of water on handgrip strength, which may lead to some bias in the research results. (3) This study used handgrip strength as an indicator of muscle strength, which is an important indicator reflecting upper limb muscle function and the physical health status of individuals. However, handgrip strength cannot fully represent whole-body muscle strength or muscle strength of other parts, such as lower limb muscle strength, trunk muscle strength, etc. Therefore, due to data source limitations, this study did not examine the impact of total water intake on muscle strength of other parts or the whole body, which may underestimate or overestimate the effect of water intake on muscle function and health. (4) This study used the NHANES data, which is a representative sample survey data based on non-institutionalized residents in the United States. However, the U.S. population has diversity and heterogeneity. Different races, regions, cultures, lifestyles, and other factors may affect the relationship between total water intake and handgrip strength level. Therefore, the results of this study cannot be directly generalized to other countries or regions’ populations. More cross-national or cross-regional comparative studies are needed. For example, different countries or regions’ climates, water sources, dietary habits, physical activity, and other factors may affect people’s water intake and drinking needs. In addition, different countries or regions’ sanitary conditions, medical resources, social security, and other factors may affect people’s handgrip strength levels and health outcomes, thus affecting the relationship between water intake and handgrip strength level. Therefore, in order to better understand the relationship between total water intake and handgrip strength level, it is necessary to consider the special situations and differences of different countries or regions and make corresponding adjustments and comparisons.

## 5. Conclusions

This study revealed a nonlinear association between daily total intake of water and handgrip strength and a gender difference between this association. The study suggested that adequate hydration may benefit muscle strength improvement and maintenance; however, excessive hydration may have adverse effects. This finding provided new clues and directions for exploring the mechanism of total water intake on muscle function and physical health, as well as new evidence and references for the management of hydration among adults.

## Figures and Tables

**Figure 1 nutrients-15-04477-f001:**
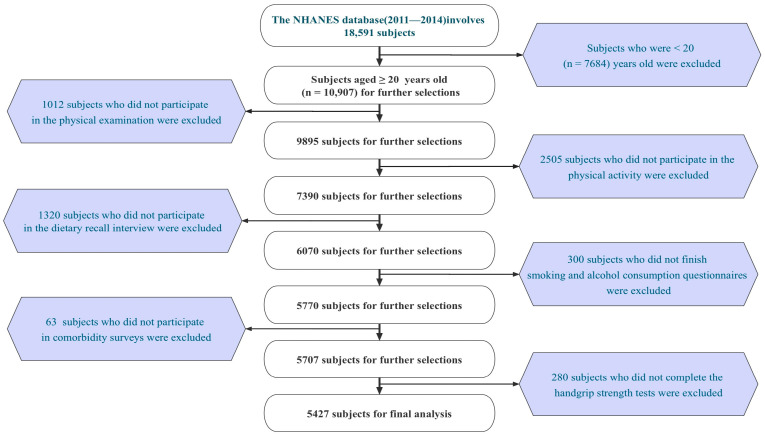
Flow chart.

**Figure 2 nutrients-15-04477-f002:**
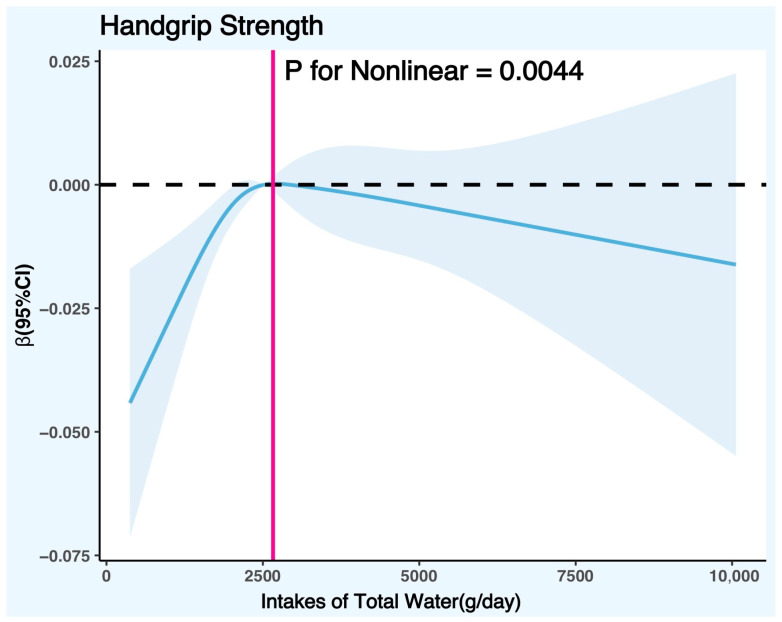
Restricted cubic spline plot model. Legend: the adjusted restricted cubic spline plot model shows an association between DTIW and HGS among all participants. The model was adjusted for gender, age, race, BMI, education level, PIR, smoking status, alcohol consumption, physical activity level, daily total intake of protein, daily total intake of energy, daily total intake of carbohydrates, daily total intake of sugars, daily total intake of fat, diabetes, hypertension, hyperlipidemia, and cancer. The blue solid line and the blue shaded area represent the estimated regression coefficient Beta and its 95% confidence interval.

**Figure 3 nutrients-15-04477-f003:**
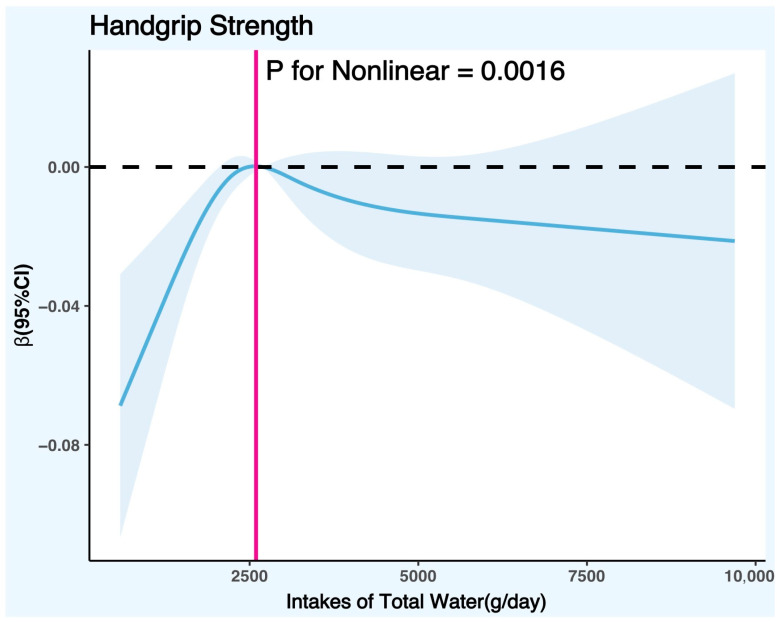
Restricted cubic spline plot model (males). Legend: the adjusted restricted cubic spline plot model shows all participants’ association between DTIW and HGS. The model was adjusted for age, race, BMI, education level, PIR, smoking status, alcohol consumption, physical activity level, daily total intake of protein, daily total intake of energy, daily total intake of carbohydrates, daily total intake of sugars, daily total intake of fat, diabetes, hypertension, hyperlipidemia, and cancer. The blue solid line and the blue shaded area represent the estimated regression coefficient Beta and its 95% confidence interval.

**Figure 4 nutrients-15-04477-f004:**
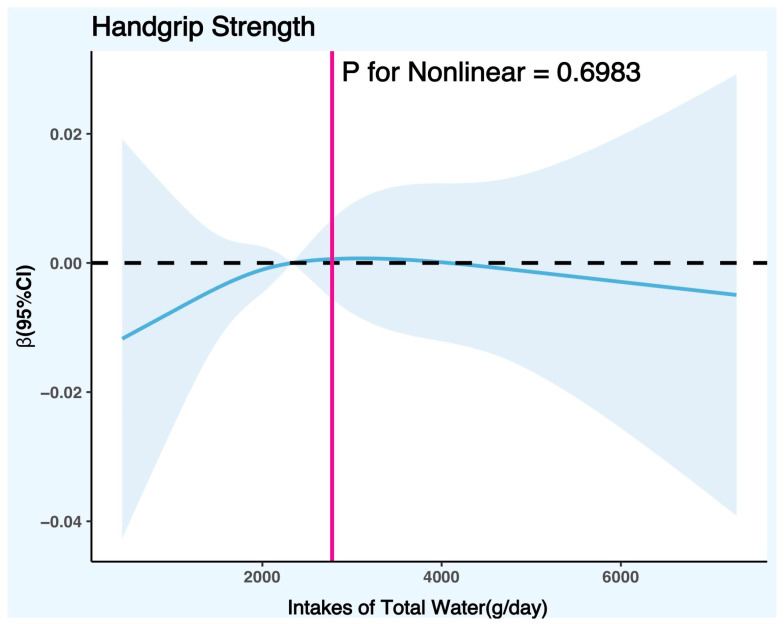
Restricted cubic spline plot model (females). Legend: the adjusted restricted cubic spline plot model shows all participants’ association between DTIW and HGS. The model was adjusted for age, race, BMI, education level, PIR, smoking status, alcohol consumption, physical activity level, daily total intake of protein, daily total intake of energy, daily total intake of carbohydrates, daily total intake of sugars, daily total intake of fat, diabetes, hypertension, hyperlipidemia, and cancer. The blue solid line and the blue shaded area represent the estimated regression coefficient Beta and its 95% confidence interval.

**Table 1 nutrients-15-04477-t001:** Baseline Characteristics of the Study Population.

Characteristic	Overall	Quartile 1 [6.93, 1855.685]	Quartile 2 (1855.685, 2517.15]	Quartile 3 (2517.15, 3358.675]	Quartile 4 (3358.675, 15,829.6]	*p*-Value
N	5427	1357	1357	1356	1357	
Handgrip Strength, kg/weight (kg)	0.93 (0.01)	0.90 (0.01)	0.91 (0.01)	0.93 (0.01)	0.97 (0.01)	<0.0001
Gender, n (weighted %)						<0.0001
Female	2577 (48.37)	760 (58.28)	715 (54.20)	623 (47.19)	479 (37.62)	
Male	2850 (51.63)	597 (41.72)	642 (45.80)	733 (52.81)	878 (62.38)	
Age, years, n (weighted %)						<0.0001
20–29	1089 (20.06)	330 (26.32)	251 (18.91)	219 (15.82)	289 (21.39)	
30–39	993 (18.29)	216 (15.65)	215 (14.46)	265 (18.57)	297 (19.42)	
40–49	974 (17.94)	201 (16.25)	248 (18.83)	251 (19.16)	274 (21.13)	
50–59	889 (16.38)	177 (16.07)	229 (19.55)	236 (20.67)	247 (21.24)	
≥60	1483 (27.32)	433 (25.72)	414 (28.26)	386 (25.78)	250 (16.83)	
Race, n (weighted %)						<0.0001
Non-Hispanic Black	1211 (10.21)	433 (18.27)	328 (11.29)	242 (7.63)	208 (6.04)	
Mexican American	567 (7.21)	133 (8.45)	145 (7.17)	136 (6.06)	153 (7.43)	
Non-Hispanic White	2406 (70.76)	464 (57.92)	594 (70.77)	666 (75.70)	682 (75.18)	
Other race (including multi-racial and other Hispanic)	1243 (11.82)	327 (15.36)	290 (10.78)	312 (10.61)	314 (11.35)	
Body mass index, kg/m^2^, n (weighted %)						0.08
<25	1713 (31.19)	445 (32.52)	447 (32.75)	430 (32.53)	391 (27.71)	
25–29.9	1800 (34.96)	440 (35.48)	413 (32.73)	488 (35.88)	459 (35.63)	
≥30	1914 (33.85)	472 (32.00)	497 (34.52)	438 (31.59)	507 (36.66)	
Education level, n (weighted %)						<0.0001
Above	863 (11.14)	281 (14.64)	220 (11.62)	189 (10.29)	173 (9.06)	
High school	1126 (19.50)	313 (23.37)	288 (19.71)	257 (18.07)	268 (17.95)	
Below	3438 (69.36)	763 (61.99)	849 (68.66)	910 (71.64)	916 (73.00)	
Poverty to income ratio, n (weighted %)						<0.0001
<1.3	1642 (20.53)	526 (30.80)	394 (19.34)	349 (17.16)	373 (17.47)	
1.3–3.49	1882 (34.33)	458 (34.11)	491 (35.80)	472 (34.19)	461 (33.36)	
≥3.5	1903 (45.14)	373 (35.09)	472 (44.86)	535 (48.65)	523 (49.16)	
Smoking status, n (weighted %)						0.002
Former smoker	1268 (24.43)	250 (18.52)	304 (24.27)	353 (26.46)	361 (26.83)	
Non-smoker	3126 (57.57)	838 (61.35)	818 (59.82)	766 (57.56)	704 (53.06)	
Current smoker	1033 (17.99)	269 (20.13)	235 (15.91)	237 (15.98)	292 (20.11)	
Alcohol status, n (weighted %)						<0.0001
Former	821 (12.56)	240 (15.40)	220 (13.60)	186 (11.73)	175 (10.45)	
Never	666 (9.51)	220 (13.86)	182 (10.95)	146 (7.65)	118 (6.98)	
Mild	1947 (37.55)	454 (32.26)	499 (39.17)	560 (44.96)	434 (33.04)	
Moderate	904 (18.95)	208 (18.07)	212 (17.23)	224 (17.99)	260 (21.88)	
Heavy	1089 (21.44)	235 (20.42)	244 (19.04)	240 (17.66)	370 (27.66)	
Physical activity, MET min/wk, n (weighted %)						<0.001
Q1 [40, 800]	1373 (23.77)	400 (28.06)	390 (27.34)	312 (21.86)	271 (19.53)	
Q2 (800, 1920]	1341 (25.16)	328 (23.37)	330 (27.41)	363 (26.76)	320 (23.03)	
Q3 (1920, 5040]	1378 (26.09)	323 (25.65)	351 (24.30)	349 (25.75)	355 (28.20)	
Q4 (5040, 59,040]	1335 (24.99)	306 (22.92)	286 (20.95)	332 (25.62)	411 (29.24)	
Daily total intake of protein, g/day, n (weighted %)						<0.0001
Q1 [0, 53.57]	1358 (23.48)	602 (44.12)	338 (23.95)	246 (17.86)	172 (13.83)	
Q2 (53.57, 75.575]	1356 (24.65)	400 (28.96)	380 (27.58)	316 (24.48)	260 (19.33)	
Q3 (75.575, 102.52]	1357 (26.58)	227 (17.84)	371 (29.89)	416 (31.28)	343 (25.58)	
Q4 (102.52, 474.19]	1356 (25.29)	128 (9.08)	268 (18.58)	378 (26.39)	582 (41.26)	
Daily total intake of energy, kcal/day, n (weighted %)						<0.0001
Q1 [93, 1498.5]	1357 (22.43)	479 (33.03)	354 (22.48)	301 (20.12)	223 (17.09)	
Q2 (1498.5, 2007]	1358 (24.93)	356 (26.41)	368 (28.69)	352 (26.61)	282 (19.18)	
Q3 (2007, 2688.5]	1355 (26.13)	287 (21.73)	359 (27.69)	351 (26.61)	358 (27.44)	
Q4 (2688.5, 12,108]	1357 (26.52)	235 (18.82)	276 (21.14)	352 (26.66)	494 (36.29)	
Daily total intake of carbohydrates, g/day, n (weighted %)						<0.0001
Q1 [3.8, 175.005]	1357 (23.87)	444 (30.74)	338 (24.69)	311 (22.14)	264 (19.98)	
Q2 (175.005, 242.16]	1356 (24.58)	348 (25.73)	356 (25.87)	357 (26.23)	295 (21.16)	
Q3 (242.16, 326.935]	1357 (26.04)	323 (24.11)	374 (28.50)	322 (25.62)	338 (25.72)	
Q4 (326.935, 1362.55]	1357 (25.51)	242 (19.43)	289 (20.94)	366 (26.01)	460 (33.14)	
Daily total intake of sugars, g/day, n (weighted %)						0.07
Q1 [0.1, 61.08]	1357 (24.34)	382 (25.28)	348 (25.98)	314 (23.89)	313 (22.72)	
Q2 (61.08, 99.43]	1357 (24.73)	343 (25.24)	331 (24.28)	367 (27.06)	316 (22.61)	
Q3 (99.43, 149.34]	1356 (25.25)	354 (26.17)	360 (26.14)	334 (25.00)	308 (24.11)	
Q4 (149.34, 1048.48]	1357 (25.68)	278 (23.32)	318 (23.60)	341 (24.05)	420 (30.57)	
Daily total intake of fat, g/day, n (weighted %)						<0.0001
Q1 [0.33, 50]	1357 (22.37)	437 (29.39)	353 (22.47)	308 (20.57)	259 (19.03)	
Q2 (50, 74.58]	1356 (24.59)	377 (28.66)	354 (26.01)	338 (23.96)	287 (21.13)	
Q3 (74.58,105.715]	1357 (26.87)	286 (22.34)	369 (30.06)	347 (28.80)	355 (25.59)	
Q4 (105.715,548.38]	1357 (26.17)	257 (19.61)	281 (21.46)	363 (26.67)	456 (34.25)	
Diabetes, n (weighted %)						0.55
No	4617 (88.90)	1121 (88.10)	1166 (88.65)	1158 (88.67)	1172 (89.87)	
Yes	810 (11.10)	236 (11.90)	191 (11.35)	198 (11.33)	185 (10.13)	
Hypertension, n (weighted %)						0.62
No	3334 (64.85)	812 (63.38)	807 (64.06)	838 (64.93)	877 (66.45)	
Yes	2093 (35.15)	545 (36.62)	550 (35.94)	518 (35.07)	480 (33.55)	
Hyperlipidemia, n (weighted %)						0.02
No	1814 (32.45)	480 (36.00)	417 (29.25)	432 (30.42)	485 (34.52)	
Yes	3613 (67.55)	877 (64.00)	940 (70.75)	924 (69.58)	872 (65.48)	
Cancer, n (weighted %)						0.05
No	4951 (89.93)	1252 (92.47)	1228 (89.02)	1211 (87.97)	1260 (90.72)	
Yes	476 (10.07)	105 (7.53)	129 (10.98)	145 (12.03)	97 (9.28)	

The continuity variables involved in this study are expressed as means (standard error), and the categorical variables are expressed as actual quantities (weighted percentages); the one-way ANOVA applies to continuity variables, and the chi-squared test applies to categorical variables; abbreviations: Q1, the first quartile; Q2, the second quartile; Q3, the third quartile; Q4, the fourth quartile; ref, reference; daily total intake of water quartile ranges: Quartile 1 = 6.93 to 1855.685 g/day; Quartile 2 = 1855.686 to 2517.15 g/day; Quartile 3 = 2517.16 to 3358.675 g/day; Quartile 4 = 3358.676 to 15,829.6 g/day.

**Table 2 nutrients-15-04477-t002:** Association Analysis between Daily Total Intake of Water and Handgrip Strength in American Adults.

	Crude Model		Model 1		Model 2	
Daily Total Intake of Water (g/day)	95%CI	*p*	95%CI	*p*	95%CI	*p*
Q1	ref		ref		ref	
Q2	0.01 (−0.02, 0.03)	0.63	0.01 (−0.01, 0.02)	0.93	0.01 (−0.01, 0.02)	0.48
Q3	0.03 (0.01, 0.05)	0.01	0 (−0.02, 0.02)	0.17	0 (−0.02, 0.02)	0.87
Q4	0.07 (0.04, 0.09)	<0.0001	0.01 (−0.01, 0.03)	0.46	0.01 (0.00, 0.03)	0.15
*p* for trend		<0.0001		0.25		0.21

Crude Model is the unadjusted model. Model 1 is adjusted for age, gender, race, poverty to income ratio, education level, BMI, physical activity, daily total intake of protein, daily total intake of energy, daily total intake of carbohydrates, daily total intake of sugars, daily total intake of fat, smoke status, and alcohol status. Model 2 is adjusted for age, gender, race, poverty to income ratio, education level, BMI, physical activity, daily total intake of protein, daily total intake of energy, daily total intake of carbohydrates, daily total intake of sugars, daily total intake of fat, smoke status, alcohol status, diabetes, hypertension, hyperlipidemia, and cancer. Abbreviations: Q1, the first quartile; Q2, the second quartile; Q3, the third quartile; Q4, the fourth quartile; ref, reference.

**Table 3 nutrients-15-04477-t003:** Association Analysis between Daily Total Intake of Water and Handgrip Strength in American Adults (Gender subgroup).

Gender: Male	Crude Model		Model 1		Model 2	
Daily total intake of water (g/day)	95%CI	*p*	95%CI	*p*	95%CI	*p*
Q1	ref		ref		ref	
Q2	0.02 (−0.02, 0.05)	0.34	0.01 (−0.01, 0.04)	0.34	0.01 (−0.01, 0.04)	0.31
Q3	0.01 (−0.02, 0.05)	0.44	0.01 (−0.01, 0.04)	0.28	0.02 (−0.01, 0.04)	0.27
Q4	0.03 (−0.01, 0.07)	0.10	0.02 (−0.01, 0.05)	0.15	0.02 (−0.01, 0.05)	0.13
*p* for trend		0.13		0.16		0.15
**Gender: Female**	**Crude Model**		**Model 1**		**Model 2**	
Daily total intake of water (g/day)	95%CI	*p*	95%CI	*p*	95%CI	*p*
Q1	ref		ref		ref	
Q2	−0.02 (−0.05, 0.00)	0.08	0 (−0.02, 0.01)	0.77	0 (−0.02, 0.01)	0.62
Q3	−0.01 (−0.03, 0.01)	0.48	−0.01 (−0.03, 0.01)	0.20	−0.01 (−0.03, 0.01)	0.25
Q4	−0.01 (−0.03, 0.01)	0.42	0 (−0.02, 0.02)	0.95	0 (−0.02, 0.02)	0.99
*p* for trend		0.71		0.65		0.77

Crude Model is the unadjusted model. Model 1 adjusted for age, race, poverty to income ratio, education level, BMI, physical activity, daily total intake of protein, daily total intake of energy, daily total intake of carbohydrates, daily total intake of sugars, daily total intake of fat, smoke status, and alcohol status. Model 2 adjusted for age, race, poverty to income ratio, education level, BMI, physical activity, daily total intake of protein, daily total intake of energy, daily total intake of carbohydrates, daily total intake of sugars, daily total intake of fat, smoke status, alcohol status, diabetes, hypertension, hyperlipidemia, and cancer. Abbreviations: Q1, the first quartile; Q2, the second quartile; Q3, the third quartile; Q4, the fourth quartile; ref, reference.

**Table 4 nutrients-15-04477-t004:** Interaction Effect Test.

	Q1	Q2	*p*	Q3	*p*	Q4	*p*	*p* for Trend	*p* for Interaction
Gender									0.2
Female	ref	0 (−0.02, 0.01)	0.62	−0.01 (−0.03, 0.01)	0.25	0 (−0.02, 0.02)	0.99	0.59	
Male	ref	0.01 (−0.02, 0.04)	0.48	0.01 (−0.01, 0.04)	0.31	0.02 (−0.01, 0.05)	0.22	0.69	
Age									0.44
20–29	ref	−0.01 (−0.05, 0.02)	0.49	−0.01 (−0.06, 0.03)	0.51	0.01 (−0.03, 0.05)	0.62	0.53	
30–39	ref	−0.01 (−0.04, 0.01)	0.3	−0.01 (−0.03, 0.02)	0.61	0 (−0.04, 0.03)	0.94	0.46	
40–49	ref	0.01 (−0.02, 0.04)	0.54	0.01 (−0.02, 0.04)	0.44	0.02 (−0.01, 0.06)	0.21	0.5	
50–59	ref	−0.01 (−0.06, 0.04)	0.72	−0.04 (−0.08, 0.01)	0.15	−0.03 (−0.07, 0.02)	0.22	0.21	
≥60	ref	0.02 (0.00, 0.05)	0.07	0.02 (−0.01, 0.05)	0.16	0.03 (0.00, 0.06)	0.09	0.6	
Race									0.24
Non-Hispanic Black	ref	0 (−0.03, 0.03)	0.82	0 (−0.03, 0.04)	0.92	0 (−0.05, 0.04)	0.87	0.81	
Mexican American	ref	0.03 (−0.02, 0.08)	0.23	0.02 (−0.03, 0.07)	0.47	0.01 (−0.04, 0.05)	0.81	0.91	
Non-Hispanic White	ref	0 (−0.02, 0.01)	0.62	−0.01 (−0.03, 0.01)	0.42	0.01 (−0.01, 0.03)	0.32	0.18	
Other race	ref	0.03 (0.00, 0.06)	0.03	0.03 (0.00, 0.07)	0.07	0.02 (−0.03, 0.06)	0.41	0.73	
Body mass index									0.18
<25	ref	−0.01 (−0.04, 0.02)	0.57	0 (−0.04, 0.04)	0.94	0.01 (−0.02, 0.05)	0.46	0.34	
25–29.9	ref	0.01 (−0.01, 0.04)	0.27	0 (−0.03, 0.02)	0.74	0.02 (0.00, 0.04)	0.1	0.24	
≥30	ref	0 (−0.02, 0.03)	0.67	0 (−0.02, 0.03)	0.75	0 (−0.02, 0.02)	0.94	0.68	
Education level									0.54
Above	ref	0.01 (−0.03, 0.06)	0.54	0.02 (−0.02, 0.06)	0.32	0.02 (−0.03, 0.07)	0.53	0.9	
High school	ref	0.02 (−0.01, 0.05)	0.19	0.01 (−0.03, 0.04)	0.68	0.03 (0.00, 0.06)	0.03	0.08	
Below	ref	−0.01 (−0.03, 0.02)	0.6	−0.01 (−0.03, 0.02)	0.68	0 (−0.02, 0.02)	0.88	0.81	
Poverty to income ratio									0.39
<1.3	ref	0.03 (0.00, 0.06)	0.07	0.02 (−0.01, 0.05)	0.12	0.03 (0.00, 0.05)	0.06	0.73	
1.3–3.49	ref	−0.01 (−0.04, 0.02)	0.44	−0.01 (−0.03, 0.02)	0.56	0.02 (−0.01, 0.05)	0.22	0.06	
≥3.5	ref	0 (−0.02, 0.03)	0.9	−0.01 (−0.03, 0.02)	0.72	−0.01 (−0.03, 0.02)	0.68	0.74	
Smoke status									0.23
Former smoker	ref	0 (−0.02, 0.02)	0.68	0 (−0.02, 0.02)	0.87	0.01 (−0.01, 0.03)	0.43	0.67	
Non-smoker	ref	0.02 (−0.01, 0.05)	0.24	0.03 (0.00, 0.06)	0.03	0.04 (0.01, 0.06)	0.01	0.14	
Current smoker	ref	−0.01 (−0.05, 0.04)	0.71	−0.02 (−0.07, 0.02)	0.26	0 (−0.04, 0.04)	0.99	0.19	
Alcohol status									0.15
Former	ref	0 (−0.04, 0.03)	0.75	0.02 (−0.02, 0.06)	0.25	0.03 (−0.01, 0.06)	0.11	0.94	
Never	ref	0.02 (−0.01, 0.05)	0.18	0 (−0.03, 0.04)	0.8	0.04 (0.01, 0.07)	0.02	0.1	
Mild	ref	0.02 (−0.01, 0.06)	0.16	0 (−0.03, 0.03)	0.78	0.01 (−0.02, 0.04)	0.58	0.13	
Moderate	ref	−0.03 (−0.07, 0.01)	0.09	−0.03 (−0.06, 0.00)	0.07	−0.01 (−0.05, 0.03)	0.47	0.26	
Heavy	ref	0.01 (−0.03, 0.05)	0.54	0.02 (−0.02, 0.07)	0.32	0.02 (−0.02, 0.06)	0.33	0.76	
Physical activity									0.09
Q1	ref	0.04 (0.01, 0.07)	0.02	0.01 (−0.02, 0.03)	0.67	0.03 (0.00, 0.06)	0.08	0.02	
Q2	ref	−0.01 (−0.04, 0.01)	0.25	−0.02 (−0.05, 0.01)	0.21	−0.03 (−0.06, 0.01)	0.1	0.38	
Q3	ref	0.01 (−0.02, 0.04)	0.6	0 (−0.04, 0.03)	0.98	0.01 (−0.02, 0.04)	0.51	0.74	
Q4	ref	−0.01 (−0.05, 0.03)	0.67	0.02 (−0.02, 0.06)	0.3	0.02 (−0.02, 0.06)	0.27	0.78	
Daily total intake of protein									0.65
Q1	ref	−0.01 (−0.03, 0.02)	0.65	0 (−0.03, 0.03)	0.99	0 (−0.03, 0.03)	0.95	1	
Q2	ref	0.02 (−0.01, 0.04)	0.27	0.02 (−0.01, 0.06)	0.17	0.02 (−0.01, 0.05)	0.16	0.93	
Q3	ref	0 (−0.03, 0.04)	0.82	−0.02 (−0.06, 0.03)	0.44	−0.01 (−0.04, 0.03)	0.68	0.71	
Q4	ref	−0.03 (−0.08, 0.01)	0.11	−0.02 (−0.07, 0.03)	0.4	−0.01 (−0.05, 0.04)	0.75	0.38	
Daily total intake of energy									0.57
Q1	ref	0 (−0.03, 0.03)	0.81	0.01 (−0.03, 0.04)	0.76	0.01 (−0.02, 0.04)	0.43	0.39	
Q2	ref	0.01 (−0.03, 0.04)	0.64	−0.02 (−0.05, 0.01)	0.3	−0.02 (−0.05, 0.02)	0.37	0.21	
Q3	ref	0.02 (−0.01, 0.05)	0.16	0.01 (−0.02, 0.05)	0.48	0.03 (0.00, 0.07)	0.05	0.09	
Q4	ref	−0.01 (−0.04, 0.02)	0.48	0.01 (−0.02, 0.05)	0.53	0.02 (−0.02, 0.05)	0.42	0.02	
Daily total intake of carbohydrates									0.65
Q1	ref	−0.01 (−0.04, 0.01)	0.31	0 (−0.03, 0.03)	0.76	−0.01 (−0.04, 0.03)	0.72	0.56	
Q2	ref	0.02 (−0.01, 0.05)	0.29	0 (−0.03, 0.03)	0.95	0.02 (−0.01, 0.05)	0.24	0.17	
Q3	ref	0.01 (−0.02, 0.04)	0.68	−0.01 (−0.03, 0.02)	0.53	0 (−0.03, 0.03)	0.95	0.22	
Q4	ref	0.01 (−0.03, 0.05)	0.61	0.02 (−0.02, 0.06)	0.29	0.03 (−0.01, 0.07)	0.18	0.14	
Daily total intake of sugars									0.77
Q1	ref	0 (−0.03, 0.03)	0.97	−0.02 (−0.05, 0.02)	0.29	0.01 (−0.03, 0.05)	0.62	0.06	
Q2	ref	0.01 (−0.03, 0.04)	0.73	−0.01 (−0.03, 0.02)	0.67	−0.01 (−0.05, 0.03)	0.48	0.53	
Q3	ref	0.01 (−0.02, 0.03)	0.6	0.01 (−0.02, 0.03)	0.57	0.01 (−0.02, 0.04)	0.41	0.79	
Q4	ref	0.01 (−0.03, 0.05)	0.62	0.02 (−0.03, 0.06)	0.39	0.03 (−0.02, 0.07)	0.22	0.73	
Daily total intake of fat									0.08
Q1	ref	0.01 (−0.02, 0.04)	0.34	0.01 (−0.02, 0.03)	0.61	0.03 (0.00, 0.06)	0.04	0.08	
Q2	ref	0.02 (−0.02, 0.05)	0.34	−0.02 (−0.05, 0.02)	0.31	0.01 (−0.03, 0.04)	0.79	0.55	
Q3	ref	−0.01 (−0.04, 0.03)	0.7	−0.01 (−0.04, 0.02)	0.6	−0.02 (−0.05, 0.02)	0.33	0.68	
Q4	ref	−0.01 (−0.04, 0.02)	0.54	0.01 (−0.02, 0.05)	0.44	0.02 (−0.01, 0.05)	0.19	0.61	
Diabetes									0.5
No	ref	0 (−0.02, 0.02)	0.8	0 (−0.02, 0.02)	0.9	0.01 (−0.01, 0.03)	0.26	0.37	
Yes	ref	0.02 (0.00, 0.04)	0.12	0.01 (−0.03, 0.05)	0.57	0.01 (−0.02, 0.04)	0.38	0.92	
Hypertension									0.44
No	ref	−0.01 (−0.03, 0.01)	0.53	0 (−0.03, 0.02)	0.8	0.01 (−0.02, 0.03)	0.65	0.96	
Yes	ref	0.02 (−0.01, 0.06)	0.19	0.01 (−0.02, 0.04)	0.67	0.02 (−0.01, 0.05)	0.22	0.3	
Hyperlipidemia									0.83
No	ref	0.01 (−0.01, 0.04)	0.36	0.01 (−0.01, 0.04)	0.33	0.02 (−0.01, 0.05)	0.24	0.7	
Yes	ref	0 (−0.02, 0.02)	0.82	−0.01 (−0.03, 0.01)	0.52	0.01 (−0.01, 0.02)	0.51	0.58	
Cancer									0.53
No	ref	0 (−0.01, 0.02)	0.57	0 (−0.02, 0.02)	0.85	0.01 (−0.01, 0.03)	0.28	0.67	
Yes	ref	0 (−0.04, 0.04)	0.9	0.03 (−0.02, 0.08)	0.21	0.03 (0.00, 0.07)	0.07	0.04	

Adjusted for age, race, poverty to income ratio, education level, BMI, physical activity, daily total intake of protein, daily total intake of energy, daily total intake of carbohydrates, daily total intake of sugars, daily total intake of fat, smoke status, alcohol status, diabetes, hypertension, hyperlipidemia, and cancer. Abbreviations: Q1, the first quartile; Q2, the second quartile; Q3, the third quartile; Q4, the fourth quartile; ref, reference.

## Data Availability

The data utilized in this study were sourced from the National Health and Nutrition Examination Survey (NHANES) for the years 2011–2014. Data for the years 2011–2012 were obtained from the NHANES website (https://wwwn.cdc.gov/nchs/nhanes/continuousnhanes/default.aspx?BeginYear=2011), while data for the years 2013–2014 were acquired from the NHANES website (https://wwwn.cdc.gov/nchs/nhanes/continuousnhanes/default.aspx?BeginYear=2013). The availability and accessibility of these datasets were ensured through the NHANES data repository.

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
