# Peer review of "Water Intake and Handgrip Strength in US Adults: A Cross-Sectional Study Based on NHANES 2011–2014 Data"

_nutrients, 2023, doi:10.3390/nu15204477_

Round 1

Reviewer 1 Report

General comments

This study aimed to examine the relationship between daily intake total of water (DITW) and handgrip strength (HGS) among US adults. It was demonstrated that there was a significant nonlinear trend between DITW and HGS, with a cut-off point of 2614.202 g/day. However, gender subgroup analysis showed that the non linear trend was significant only for males, with a cut-off point of 2549.295 25 g/day. None of the stratified variables had an interaction effect on the association between DITW and HGS. The authors concluded that these findings provide new clues and directions for exploring the mechanism of the impact of DITW on muscle function and health, and also provide new evidence and suggestions for adults to adjust their water intake reasonably. Results are novel. The article is well written and the topic interesting and relevant. Below I provide comments for authors to consider.

 Abstract

 1.     It would be interesting to provide a one-sentence rationale.

2.     Line 17, please report the percentage of men and women who participated in the study.

3.     Line 23, please explain the nature of the relationship. It is important. The same should apply for line 25.

4.     Line 24, please remove the decimals from the number. The same should apply for line 25.

5.     Line 27, In conclusion, this study found….

 Introduction

 The introduction is clear and provide sufficient information for the readers to understand the need and rationale of the study.

 Line 66. A study by Goulet et al. would be relevant here.

 Goulet EDB, Mélançon MO, Lafrenière D, Paquin J, Maltais M, Morais JA. Impact of Mild Hypohydration on Muscle Endurance, Power, and Strength in Healthy, Active Older Men. J Strength Cond Res. 2018 Dec;32(12):3405-3415. doi: 10.1519/JSC.0000000000001857. PMID: 28234715.

 Line 74. Should it not be the other way around, i.e., muscle strength loss caused by muscle mass loss.

 Methods

 Figure 1. I cannot understand with this diagram how you arrived at a number of 5428 individuals. The flow is not intuitive.

 Line 113. Please provide a reference for the use of a two non-consecutive day dietary intake questionnaire.

 Results 

This section is fine.

 Discussion

Lines 359-403. These paragraphs are interesting, and I understand that you are trying to speculate. However, some caution needs to be used here to ensure that things that are written are in line with our current knowledge of physiology and/or are supported by appropriate research findings. Also, care must be taken for not oversimplifying the possible relationships or explanations. For instance, line 377, …the circulation is blocked. This does not make sense in the context of this study, and this is an oversimplification in that the role of vessels dilatation is omitted. Another example is that lines 389 to 404 are unsupported by relevant research findings.

Line 425. The fact that a dietary recall was used to assess water intake also represents a limitation of the study.

The article is well written.

Author Response

Reviewer 1

Thank you for the valuable recommendations provided by the reviewing experts for this study. Your suggestions are of significant importance in enhancing the quality of this manuscript. Our team has incorporated your feedback by making revisions and providing a comprehensive description in the discussion section. In consideration of your advice, we have also removed unnecessary content. However, we have appropriately supplemented the discussion section based on feedback from other reviewers. We have addressed grammar and punctuation issues found in the original manuscript. We hope for your understanding. Our team greatly appreciates your detailed feedback on every aspect, and the following is our specific response to your recommendations:

  • Line 17, please report the percentage of men and women who participated in the study.

This has been modified in accordance with expert recommendations.

Revised: Using data from the National Health and Nutrition Examination Survey (NHANES) 2011-2014, a cross-sectional survey design was adopted to analyze 5427 adults(48.37% female and 51.63% male) aged 20 years and above.

  • Line 23, please explain the nature of the relationship. It is important. The same should apply for line 25.

This has been modified in accordance with expert recommendations.

Revised: There was a significant nonlinear trend(exhibiting an inverted U-Curve) between DITW and HGS (P for Nonlinear=0.0044), with a cut-off point of 2663 g/day. Gender subgroup analysis showed that the nonlinear trend(exhibiting an inverted U-Curve) was significant only for males (P for Nonline-ar=0.0016), with a cut-off point of 2595 g/day. None of the stratified variables had an interaction effect on the association between DITW and HGS (P for interaction>0.05)

  • Line 24, please remove the decimals from the number. The same should apply for line 25.

This has been modified in accordance with expert recommendations.

Revised: There was a significant nonlinear trend(exhibiting an inverted U-Curve) between DITW and HGS (P for Nonlinear=0.0044), with a cut-off point of 2663 g/day. Gender subgroup analysis showed that the nonlinear trend(exhibiting an inverted U-Curve) was significant only for males (P for Nonline-ar=0.0016), with a cut-off point of 2595 g/day

  • Line 27, In conclusion, this study found….

This has been modified in accordance with expert recommendations.

Revised: In conclusion, this study found a nonlinear association between DITW and HGS levels, and there was a gender difference. This finding provides new clues and directions for exploring the mechanism of the impact of DITW on muscle function and health and also provides new evidence and suggestions for adults to adjust their water intake reasonably.

  • Line 66. A study by Goulet et al. would be relevant here.

Goulet EDB, Mélançon MO, Lafrenière D, Paquin J, Maltais M, Morais JA. Impact of Mild Hypohydration on Muscle Endurance, Power, and Strength in Healthy, Active Older Men. J Strength Cond Res. 2018 Dec;32(12):3405-3415. doi: 10.1519/JSC.0000000000001857. PMID: 28234715.

This has been modified in accordance with expert recommendations.

Revised:This decline in hydration status not only predisposes the elderly to various health risks but also potentially exacerbates age-related decrements in muscle mass and functional strength, engendering a complex interplay among aging, hydration, and muscular function[9].

  1. Goulet, E.D.B.; Mélançon, M.O.; Lafrenière, D.; Paquin, J.; Maltais, M.; Morais, J.A. Impact of Mild Hypohydration on Muscle Endurance, Power, and Strength in Healthy, Active Older Men. J Strength Cond Res2018, 32, 3405–3415, doi:10.1519/JSC.0000000000001857.

  • Line 74. Should it not be the other way around, i.e., muscle strength loss caused by muscle mass loss.

This has been modified in accordance with expert recommendations.

Revised:Previous studies have predominantly concentrated on the relationship between hydration status and muscle mass, whereas it is essential to highlight that the bulk of muscle strength decline cannot be solely attributed to the reduction in muscle mass. This implies the potential existence of other factors that may exert a more substantial influence on muscle strength than mere changes in muscle mass[14][15].

  • Figure 1. I cannot understand with this diagram how you arrived at a number of 5428 individuals. The flow is not intuitive.

This has been modified in accordance with expert recommendations.

Revised:

  • Line 113. Please provide a reference for the use of a two non-consecutive day dietary intake questionnaire.

The questionnaire is placed at the end of this paper.

  • Lines 359-403. These paragraphs are interesting, and I understand that you are trying to speculate. However, some caution needs to be used here to ensure that things that are written are in line with our current knowledge of physiology and/or are supported by appropriate research findings. Also, care must be taken for not oversimplifying the possible relationships or explanations. For instance, line 377, …the circulation is blocked. This does not make sense in the context of this study, and this is an oversimplification in that the role of vessels dilatation is omitted. Another example is that lines 389 to 404 are unsupported by relevant research findings.

This has been modified in accordance with expert recommendations and supplemented by the reference.

Revised:Nervous function refers to the ability of neurons to communicate information with other cells through potential changes or neurotransmitter release[71][72]. Nervous system function is also affected by hydration status[73][74]. When the body is in a low hydration state (or dehydration), the water and electrolyte imbalance inside and outside the nerve cells leads to abnormal membrane potential and action potential of nerve cells and synaptic transmission disorder between nerve cells[75] (indirectly leading to neuro-transmitter release and binding obstruction, nerve signal transmission delay or distortion, nerve network stability and plasticity reduction and other series of adverse consequences); when the body is in a high hydration state (or overhydration), the water and electrolyte excess inside and outside the nerve cells leads to abnormal membrane potential and action potential of nerve cells, and synaptic transmission abnormality between nerve cells (indirectly leading to neurotransmitter release and binding abnormality, nerve signal transmission too fast or chaotic, nerve network stability and plasticity reduction and other series of adverse consequences). These changes will affect the excitability and conduction speed of motor neurons, which determine the speed and quality of motor commands from the brain to the muscles. When the excitability and conduction speed of motor neurons are low, it will cause motor command transmission delay or distortion, thus affecting the coordination and strength of muscle contraction[76][77].

  • Line 425. The fact that a dietary recall was used to assess water intake also represents a limitation of the study.

This has been modified in accordance with expert recommendations.

Revised:2) This study used total water intake as an indicator of water intake, which includes water from drinking water, beverages, and food. However, different sources of water may have different effects on handgrip strength levels. For example, sugary drinks may affect blood glucose and insulin levels, thus affecting muscle metabolism and synthesis; water in food may interact with other nutrients, thus affecting muscle function and health. It is im-portant to note that the use of dietary recall to assess water intake also represents a study limitation. Due to data source limitations, this study did not distinguish the effects of different sources of water on handgrip strength, which may lead to some bias in the research results.

------------------------------------------------------------------------------------------------------

We have carefully considered and incorporated every recommendation made by the reviewing experts. We hope that these modifications meet the expectations of the reviewing experts and enhance the quality and comprehensibility of the manuscript. Once again, we sincerely appreciate your thorough review, and your feedback has been instrumental in further refining and improving this paper. If you have any additional recommendations or require further information, please do not hesitate to share your insights at any time.

Reviewer 2 Report

Dear Authors,

Thank you for your work, it was an interesting read. The topic is relevant and important. However, overall the manuscript was too long, especially the discussion. The last sentence of the conclusion, indicated that recommendations for hydration management were included in the manuscript, however, these were not clear and should be listed. One caution though is that recommendations are based on one observational study. Recommendations should be the focus of the discussion - what are the key messages from the analysis?

I selected moderate English corrections needed. There are a lot of spaces missing, and some word choices could be improved to explain your message.

Line 35 - first sentence, change to "Water is an essential nutrient..."

Line 44 - you skip from body water to muscle water, pick one or provide an estimate for both.

line 68 - change to "...lower handgrip..."

Line 82 - need to explain why NHANES 2011-2014 when more recent data sets are available.

Line 85 - the term "mainly drawn" implies some participants were included from another study, please correct.

Figure 1 is not clear - needs to be in a stepwise manner.

Line 126 "water content per milliliter of selected beverages" is not explained clearly. This implies you did not include water from foods.

Line 129 - what are the "other sources" it is important to show all the water sources you used to calculate water intake. Did you account for metabolic water?

Line 129-132 - references not in correct format - was this sentence copied and pasted from somewhere?

Line 144 - small k ("k") is kilo not capital K.

Line 153 - no explanation as to why you chose protein intake. What is its relationship to water? Glucose intake would be important as well as electrolyte intake. Need to explain clearly why protein was used and why not other relevant nutrients. Also, which quartile indicates the RDA for protein?

Section 3 Results - please ensure all results are in the past tense.

Table 1 - demographic results are interesting but may just be statistically relevant as opposed to clinically relevant. Any insights into this?

Tables 2, 3 and 4 could be supplementary material.

Figures 2, 3 and 4 is there one that is the most important - add the rest as supplementary material.

Line 296 - HGS decreased with increase DITW - how clinically relevant is this? Water intoxication does exist but requires far more water than what you found here to have negative physiological effects.

Lots of small things, especially spaces, word choices. Overall good English.

Author Response

Reviewer 2

Thanks to the reviewing experts for providing valuable recommendations for this study. Your recommendations are of significant importance in enhancing and improving the quality of this manuscript. Combining your feedback, our team has made modifications, including changing the word “recommendations” to “references” in the conclusion section. We have also provided a comprehensive description in the discussion section, and in consideration of your recommendation, removed non-essential contents. However, we have supplemented the discussion section appropriately in line with recommendation from other reviewers.The improper use of grammar and punctuation in the manuscript has been revised. We hope for your understanding. Our team greatly appreciates your detailed comments on every aspect, and the following is our specific response to your recommendations(Further details can be found in the supplementary materials):

  • Line 35 - first sentence, change to "Water is an essential nutrient..."

This has been modified in accordance with expert recommendations.

Revised: Water is an essential nutrient for normal physiological and metabolic processes of the human body, and it is vital for the normal function maintenance of human organs and systems.

  • Line 44 - you skip from body water to muscle water; pick one or provide an estimate for both.

This has been modified in accordance with expert recommendations.

Revised: With advancing age, there is a notable reduction in total body water (TBW) and intra-cellular water (ICW)[8]. This decline in hydration status not only predisposes the elderly to various health risks but also potentially exacerbates age-related decrements in muscle mass and functional strength, engendering a complex interplay among aging, hydration, and muscular function[9].

  • line 68 - change to "...lower handgrip..."

This has been modified in accordance with expert recommendations.

Revised: Joana et al. [12]investigated the relationship between hydration status and handgrip strength in elderly people and found that insufficient hydration in women was significantly associated with lower handgrip strength.

  • Line 82 - need to explain why NHANES 2011-2014 when more recent data sets are available.

Revised in accordance with expert recommendations and based on the content of the article, the modifications have been placed in the handgrip strength testing section, as muscle strength testing was conducted only between 2011 and 2014.

Revised: In this study, handgrip strength was used as a dependent variable, measured in Kg. Handgrip Strength(HGS) was measured by Takei Dynamometer(TKK 5401; Takei Scientific Instruments, Tokyo, Japan) for adults aged 20 years and older. It is worth mentioning that handgrip strength measurements were only available for the years 2011 to 2014, as this was the period during which the handgrip tests were conducted as part of the survey.

Subjects were asked to maintain an upright posture, arm vertically downward, handgrip the dynamometer for strength test, both hands (dominant hand, non-dominant hand) repeated the test three times, with a 60-second interval between each measurement, and the sum of the average of the highest peak handgrip strength of both hands was taken as the maximum absolute handgrip strength. To further improve the study's objectivity, relative handgrip strength was used for subsequent analysis in this study[19][20].

  • Line 85 - the term "mainly drawn" implies some participants were included from another study; please correct.

This has been modified in accordance with expert recommendations.

Revised: The study participants were sourced from the NHANES database by the Strengthening the Reporting of Observational Studies in Epidemiology (STROBE) guidelines for cross-sectional studies.

  • Figure 1 is not clear - needs to be in a stepwise manner.

This has been modified in accordance with expert recommendations.

Revised :

  • Line 126 "water content per milliliter of selected beverages" is not explained clearly. This implies you did not include water from foods.

In accordance with expert recommendations, revisions have been made to provide a detailed description of the water intake process and circumstances.

Revised: In the NHANES study, a face-to-face 24-hour dietary recall interview was conducted with each participant during their visit to the Mobile Examination Center (MEC) within a dedicated dietary interview room. This room was equipped with essential tools, including a computer equipped with the United States Department of Agriculture (USDA) Automated Multiple Pass software, food models, and various three-dimensional measuring instruments such as glasses, bowls, mugs, measuring mounds, circles, thickness sticks, spoons, rulers, cartons, and water bottles of different sizes.MEC interviewers, who had received specialized training, elucidated the purpose of the dietary recall, elucidated the interview process in detail, and presented standardized questions to each participant in a completely unbiased manner. Initially, participants were instructed to retrospectively recall all foods and beverages consumed during the preceding 24-hour period (from midnight to midnight), encompassing items consumed both at home and away, including snacks, coffee, soft drinks, water, and alcoholic beverages. Specific inquiries were made regarding brand details, preparation methods, and quantities consumed, including queries about food additives, such as milk added to cereal or coffee. Food models and measuring tools were employed to assist participants in estimating portion sizes[16].

Regarding water intake, participants were queried about their consumption of tap water, including filtered tap water and water from drinking fountains. Additionally, various brands of bottled waters (plain, spring, mineral, and electrolyte-fortified) and carbonated plain waters (sparkling, seltzer, and club soda) were considered. Water intake also encompassed liquids added to food and beverages, such as various types of liquid milk, fruit juice, vegetable juice, juice drinks, carbonated and non-carbonated sugared beverages, coffee, tea, hot chocolate, and alcoholic beverages[17][18]. This intake was calculated by accumulating water content per milliliter of each beverage. Ice was also accounted for in the recording process. As participants reported each food and beverage, MEC interviewers entered the data into the USDA Automated Multiple Pass software, cross-referencing it with entries in the USDA nutrient composition database. The software consolidated the entered data and generated estimates for total daily nutrient intake, including category-specific nutrient intake such as water, protein, carbohydrates, and other nutrients, which were utilized in our analysis.

  • Line 129 - what are the "other sources" it is important to show all the water sources you used to calculate water intake. Did you account for metabolic water?

In accordance with expert recommendations, revisions have been made to provide a detailed description of the water intake process and circumstances.

Revised: In the NHANES study, a face-to-face 24-hour dietary recall interview was conducted with each participant during their visit to the Mobile Examination Center (MEC) within a dedicated dietary interview room. This room was equipped with essential tools, including a computer equipped with the United States Department of Agriculture (USDA) Automated Multiple Pass software, food models, and various three-dimensional measuring instruments such as glasses, bowls, mugs, measuring mounds, circles, thickness sticks, spoons, rulers, cartons, and water bottles of different sizes.MEC interviewers, who had received specialized training, elucidated the purpose of the dietary recall, elucidated the interview process in detail, and presented standardized questions to each participant in a completely unbiased manner. Initially, participants were instructed to retrospectively recall all foods and beverages consumed during the preceding 24-hour period (from midnight to midnight), encompassing items consumed both at home and away, including snacks, coffee, soft drinks, water, and alcoholic beverages. Specific inquiries were made regarding brand details, preparation methods, and quantities consumed, including queries about food additives, such as milk added to cereal or coffee. Food models and measuring tools were employed to assist participants in estimating portion sizes[16].

Regarding water intake, participants were queried about their consumption of tap water, including filtered tap water and water from drinking fountains. Additionally, various brands of bottled waters (plain, spring, mineral, and electrolyte-fortified) and carbonated plain waters (sparkling, seltzer, and club soda) were considered. Water intake also encompassed liquids added to food and beverages, such as various types of liquid milk, fruit juice, vegetable juice, juice drinks, carbonated and non-carbonated sugared beverages, coffee, tea, hot chocolate, and alcoholic beverages[17][18]. This intake was calculated by accumulating water content per milliliter of each beverage. Ice was also accounted for in the recording process. As participants reported each food and beverage, MEC interviewers entered the data into the USDA Automated Multiple Pass software, cross-referencing it with entries in the USDA nutrient composition database. The software consolidated the entered data and generated estimates for total daily nutrient intake, including category-specific nutrient intake such as water, protein, carbohydrates, and other nutrients, which were utilized in our analysis.

  • Line 129-132 - references not in the correct format - was this sentence copied and pasted from somewhere?

Revised based on expert recommendations, this study's references were managed using Zotero software. There were formatting errors in the citation style within the original manuscript (Zhou et al., 2022)(Yang and Chun, 2015). These citations correspond to references 17 and 18.

  1. Ji, C.; Xia, Y.; Tong, S.; Wu, Q.; Zhao, Y. Association of Handgrip Strength with the Prevalence of Metabolic Syndrome in US Adults: The National Health and Nutrition Examination Survey. Aging (Albany NY) 2020, 12, 7818–7829, doi:10.18632/aging.103097.
  2. Bohannon, R.W.; Wang, Y.-C.; Yen, S.-C.; Grogan, K.A. Handgrip Strength: A Comparison of Values Obtained From the NHANES and NIH Toolbox Studies. Am J Occup Ther 2019, 73, 7302205080p1-7302205080p9, doi:10.5014/ajot.2019.029538.

Revised: To further improve the objectivity of the study, relative handgrip strength was used for subsequent analysis in this study[19][20]

  • Line 144 - small k ("k") is kilo, not capital K.

Revised in accordance with expert recommendations.

Revised :

Relative maximal handgrip strength = maximal handgrip strength (kg) / weight (kg)

  • Line 153 - no explanation as to why you chose protein intake. What is its relationship to water? Glucose intake would be important, as well as electrolyte intake. Need to explain clearly why protein was used and why not other relevant nutrients. Also, which quartile indicates the RDA for protein?

Revised in accordance with expert recommendations, and following a secondary discussion on this study by our research team and invited experts to enhance the objectivity of experimental results, additional covariates impacting muscle strength (daily intake of carbohydrates, sugars, fats, energy, and cancer) were included based on protein intake, along with the corresponding references.

(1) Consistency in Study Design:

This study transformed multiple nutritional and lifestyle variables, such as protein, sugars, carbohydrates, energy, fat, and physical activity into quartile groups as control variables. This approach provides a unified and consistent way to handle multiple variables in the study, simplifying the model and ensuring that all variables are treated and interpreted in a similar manner.

(2) Control for Individual Differences:

Since dietary recommendations typically depend on individual characteristics like body weight, direct use of Recommended Dietary Allowances (RDA) might not fully account for individual differences. Transforming covariates such as protein intake into quartile groups helps control for some of these individual differences to allow for a more accurate assessment of their impact.

(3) Accuracy and Robustness of Statistical Analysis:

Quartile grouping provides a clear view of data distribution while maintaining robustness to outliers, which can be an important consideration when exploring the relationship between water intake and handgrip strength.

(4) Practical Implications for Public Health Policy and Health Education:

Using quartile grouping may offer a practical approach for evaluating and comparing the effects of different nutrient intakes for public health policies and health education rather than relying solely on fixed RDA values. Combining this approach with the study's data may provide more practical and direct insights, aiding policymakers and the public in understanding the relationship between water intake and muscle function and health. Revised: In terms of the rationale for covariate selection, the scientific rigor of our study is ensured through a process of logical sorting and screening.

Primary demographic characteristics considered include age[21], gender[22][23], BMI[24], and educational level [25] due to their significant correlations with muscular strength. Additionally, research has indicated a positive correlation between higher income and superior performance across multiple domains of physical function[26]. Smoking and alcohol consumption have been demonstrated to exert noticeable negative effects on muscular composition and strength in both human[27][28] and animal studies[29][30].

Moreover, scientific evidence unequivocally underscores the beneficial impact of systematic physical activity on enhancing human health and muscular strength levels[28][31][32]. Furthermore, the intake of scientific nutrients, including protein[33], fat[34][35],carbohydrates[36][37],energy[38], and sugar[39][40], plays a pivotal role in muscular development. Existing studies have affirmed that common societal chronic conditions such as hypertension[41][42], hyperlipidemia[43][44][45], diabetes[46][47][48], and cancer[49] operate through distinct physiological pathways to mediate the decline in muscular strength and muscle mass.

  • Section 3 Results - please ensure all results are in the past tense.

Revised based on expert recommendations.

  • Table 1 - demographic results are interesting but may just be statistically relevant as opposed to clinically relevant. Any insights into this?

Revised based on expert recommendations.

Revised: This study unveiled the associations between various multifaceted factors and daily intake of total water (DITW). Individuals who were male or possessed higher education levels exhibited higher water intake, while water intake gradually decreased in adults aged over 60. Compared to other races, the water intake was highest among the white population and lowest among the black population. Economically, an upward trend in water intake was observed with the enhancement of personal economic status. Non-smokers and mild drinkers exhibited the highest water intake, displaying a positive association between healthy lifestyle choices and adequate water intake. From the perspective of nutritional intake and physical activity level, a positive correlation trend with water intake was noticed, suggesting that physical activity and appropriate nutritional intake might promote higher water intake. Compared to the healthy population, individuals with hyperlipidemia had higher water intake, and concurrently, an upward trend in water intake was observed with the increase in population handgrip strength level. These findings provided insights for social health and clinical practice, emphasizing the significance of considering multifaceted factors at both individual and community levels in public health strategies to promote adequate water intake and overall health.

  • Tables 2, 3 and 4 could be supplementary material.

Thanks for the expert recommendations. The supplementary material will be uniformly revised during the Proof stage.

  • Figures 2, 3 and 4 is there one that is the most important - add the rest as supplementary material.

Thanks for the expert recommendations. The supplementary material will be uniformly revised during the Proof stage.

  • Line 296 - HGS decreased with increase DITW - how clinically relevant is this? Water intoxication does exist but requires far more water than what you found here to have negative physiological effects.

Thank you for the expert recommendations. The expert input is indeed of significant importance for the enhancement and improvement of this manuscript. While water intoxication typically requires much higher water intake than what your study found, the association between higher DITW and decreased HGS may still have clinical relevance. Here are some considerations:

(1) Individual Variability: People vary in their water intake needs and tolerance. Some individuals may be more susceptible to the effects of excessive water intake, even if it falls within what is considered a safe range for most. Understanding these differences is crucial for personalized health recommendations.

(2) Hydration Balance: The relationship between water intake and HGS may highlight the importance of maintaining hydration balance. While excessive water intake may not lead to water intoxication, it could still impact overall hydration balance, thereby affecting muscle function and strength.

(3) Environmental Significance: The observed clinical relevance may also depend on the environment. For instance, in specific clinical populations or circumstances, even minor changes in HGS may have meaningful implications for functional capacity and quality of life.

(4) Future Research: This finding can serve as a basis for further research to explore the potential mechanisms behind the observed decrease in HGS with higher DITW. Understanding the underlying physiological processes can provide more insights into the clinical significance.

In summary, while the decrease in HGS associated with higher DITW may not directly indicate water intoxication, it does raise questions about the potential impact of hydration balance on muscle function. Future research and clinical studies can help elucidate the practical significance of this relationship and its relevance to specific populations or situations.

--------------------------------------------------------------

We have carefully considered and incorporated every recommendation made by the reviewing experts. We hope that these modifications meet the expectations of the reviewing experts and enhance the quality and comprehensibility of the manuscript. Once again, we sincerely appreciate your thorough review, and your feedback has been instrumental in further refining and improving this paper. If you have any additional recommendations or require further information, please do not hesitate to share your insights at any time.

Reviewer 3 Report

This study aimed to examine the relationship between daily intake total of water (DITW) and handgrip strength (HGS) among US adults, and to explore the impact of water intake on muscle function and health. This study is very great for readers of Nutrients. However, several changes are raised to improvement the manuscript prior the acceptance for publication.

Introduction:

-What is hypothesis?

Methods:

-Presence of chronic disease, such as cancer must be used to adjust the data.

Results and Discussion

-To add the presence of cancer (yes or not) and adjust the associations as well as a discussion of inflammatory disease on impact in hydration and dehydration.

Author Response

Reviewer 3

Thank you for the valuable recommendations provided by the reviewing experts for this study. Your suggestions are of significant importance in enhancing the quality of this manuscript. Our team has incorporated your feedback by making revisions and providing a comprehensive description in the discussion section. In consideration of your advice, we have also removed unnecessary content. However, we have appropriately supplemented the discussion section based on feedback from other reviewers. We have addressed grammar and punctuation issues found in the original manuscript. We hope for your understanding. Our team greatly appreciates your detailed feedback on every aspect, and the following is our specific response to your recommendations(Further details can be found in the supplementary materials):

  • What is hypothesis?

This has been supplied in accordance with expert recommendations.

Revised: Our study posits the existence of a potential linear or nonlinear relationship between daily total water intake (DITW) and handgrip strength (HGS) among the adult population, with the possibility of gender-based variations. Specifically, it suggests that hydration status may exacerbate age-related declines in muscle mass and functional strength.

  • Presence of chronic disease, such as cancer must be used to adjust the data.

This has been supplied in accordance with expert recommendations.

Revised: In terms of the rationale for covariate selection, the scientific rigor of our study is ensured through a process of logical sorting and screening. Primary demographic characteristics considered include age[21],gender[22][23], BMI[24], and educational level[25] due to their significant correlations with muscular strength. Additionally, research has indicated a positive correlation between higher income and superior performance across multiple domains of physical function[26]. Smoking and alcohol consumption has been demonstrated to exert noticeable negative effects on muscular composition and strength in both human[27][28] and animal studies[29][30]. Moreover, scientific evidence unequivocally underscores the beneficial impact of systematic physical activity on enhancing human health and muscular strength levels[28][31][32]. Furthermore, the intake of scientific nutrients, including protein[33], fat[34][35],carbohydrates[36][37],energy[38], and sugar[39][40], plays a pivotal role in muscular development. Existing studies have affirmed that common societal chronic conditions such as hypertension[41][42],hyperlipidemia[43][44][45], diabetes[46][47][48], and cancer[49] operate through distinct physiological pathways to mediate the decline in muscular strength and muscle mass.

Demographic factors: age, gender (male, female), race (Mexican American, non-Hispanic black, non-Hispanic white, other race), education level (Below: less than high school; High school; Above more than high school), and body mass index (<25, 25-29.9, ≥30 kg/m2).

Socioeconomic status: poverty income ratio, calculated by dividing household (or individual) income by the poverty guideline for the survey year (low income PIR≤1.3; medium income 1.3<PIR<3.5; high income≥3.5).

Dietary nutrient intake: daily intake total of protein, fat, carbohydrates, energy, and sugar recorded by two non-consecutive 24-hour dietary recall interviews and calculated based on dietary nutrient content.

Lifestyle habits: smoking status (never: smoked less than 100 cigarettes in a lifetime; former: smoked more than 100 cigarettes in lifetime, now not at all; current: smoked more than 100 cigarettes in lifetime, some days or every day), alcohol consumption (never: <12 drinks in a lifetime; former: ≥1 drink in past 12 years, none in the past year, or none in the past year but ≥12 drinks in a lifetime; mild: ≤1 drink per day for female, ≤2 drinks per day for male; moderate: ≤2 drinks per day for female, ≤3 drinks per day for males; and heavy: ≤3 drinks per day for female, ≤4 drinks per day for male), physical activity (time and energy expenditure of typical physical activity in the past week, including vigorous and moderate physical activity at work, commuting, and leisure time), measured by metabolic equivalent (weekly metabolic equivalent score, METs-h/week).

Health conditions: hypertension diagnosis (diagnosed by a doctor or health professional? ever used antihypertensive drugs? systolic blood pressure ≥140mmHg and diastolic blood pressure ≥90mmHg in three blood pressure measurements), hyperlipidemia diagnosis (triglycerides (TG) ≥150mg/dL; serum total cholesterol (TC) ≥200mg/dL; low-density lipoprotein (LDL) ≥130mg/dL; high-density lipoprotein (HDL) <40mg/dL (male), <50mg/dL (female); use of lipid-lowering drugs), diabetes diagnosis (diagnosed by a doctor or health professional? glycated hemoglobin (HbA1c) ≥6.5mmol/L; fasting blood glucose (GHLU) ≥7.0mmol/L; ever used anti-diabetic drugs?),cancer diagnosis(diagnosed by a doctor or health professional?).

  • To add the presence of cancer (yes or not) and adjust the associations as well as a discussion of inflammatory disease on impact in hydration and dehydration.

This has been supplied in accordance with expert recommendations.We have incorporated cancer as a covariate into the analysis based on your feedback. However, as this variable did not exhibit significant statistical significance in the analysis of this study, it was not emphasized in the discussion. Nevertheless, valuable information and relevant literature have been described in the supplementary materials.

Revised: This study also pays special attention to the impact of cancer on muscle strength. Specifically, cancer affects muscle strength through two primary mechanisms: first, by inducing cachexia, which leads to significant losses in body weight and muscle mass, thereby directly reducing muscle strength; and second, through the activation of the NF-κB signaling pathway[56]. This activation triggers inflammation and muscle degradation, further weakening muscle strength and disrupting the expression of Pax7, a critical factor for muscle regeneration. The cumulative effect of these factors leads to a significant decline in muscle strength among cancer patients.

------------------------------------------------------------------------------------------------------

We have carefully considered and incorporated every recommendation made by the reviewing experts. We hope that these modifications meet the expectations of the reviewing experts and enhance the quality and comprehensibility of the manuscript. Once again, we sincerely appreciate your thorough review, and your feedback has been instrumental in further refining and improving this paper. If you have any additional recommendations or require further information, please do not hesitate to share your insights at any time.

Round 2

Reviewer 2 Report

Thank you for improving the manuscript.

Few minor grammatical errors - copy editing will be able to correct this. Small "k" for "kg".